# Aprotinin Inhibits SARS-CoV-2 Replication

**DOI:** 10.3390/cells9112377

**Published:** 2020-10-30

**Authors:** Denisa Bojkova, Marco Bechtel, Katie-May McLaughlin, Jake E. McGreig, Kevin Klann, Carla Bellinghausen, Gernot Rohde, Danny Jonigk, Peter Braubach, Sandra Ciesek, Christian Münch, Mark N. Wass, Martin Michaelis, Jindrich Cinatl

**Affiliations:** 1Institute for Medical Virology, University Hospital, Goethe University, 60596 Frankfurt am Main, Germany; Denisa.Bojkova@kgu.de (D.B.); marco.bechtel@kgu.de (M.B.); Sandra.ciesek@kgu.de (S.C.); 2School of Biosciences, University of Kent, Canterbury CT2 7NJ, UK; km625@kent.ac.uk (K.-M.M.); jem53@kent.ac.uk (J.E.M.); 3Faculty of Medicine, Institute of Biochemistry II, Goethe University, 60590 Frankfurt am Main, Germany; klann@em.uni-frankfurt.de (K.K.); ch.muench@em.uni-frankfurt.de (C.M.); 4Department of Respiratory Medicine and Allergology, University Hospital, Goethe University, 60590 Frankfurt am Main, Germany; c.bellinghausen@med.uni-frankfurt.de (C.B.); Gernot.rohde@kgu.de (G.R.); 5Institute of Pathology, Hannover Medical School (MHH), 30625 Hannover, Germany; jonigk.danny@mh-hannover.de (D.J.); Braubach.Peter@mh-hannover.de (P.B.); 6Biomedical Research in Endstage and Obstructive Lung Disease Hannover (BREATH), The German Center for Lung Research (Deutsches Zentrum für Lungenforschung, DZL), Hannover Medical School (MHH), 30625 Hannover, Germany; 7German Center for Infection Research, DZIF, External Partner Site, 60596 Frankfurt am Main, Germany; 8Fraunhofer Institute for Molecular Biology and Applied Ecology (IME), Branch Translational Medicine und Pharmacology, 60596 Frankfurt am Main, Germany; 9Frankfurt Cancer Institute, Goethe University, 60596 Frankfurt am Main, Germany; 10Cardio-Pulmonary Institute, Goethe University, 60590 Frankfurt am Main, Germany

**Keywords:** severe acute respiratory syndrome coronavirus, severe acute respiratory syndrome coronavirus 2, 2019-nCoV, COVID-19, antiviral, drug discovery, aprotinin

## Abstract

Severe acute respiratory syndrome virus 2 (SARS-CoV-2) is the cause of the current coronavirus disease 19 (COVID-19) pandemic. Protease inhibitors are under consideration as virus entry inhibitors that prevent the cleavage of the coronavirus spike (S) protein by cellular proteases. Herein, we showed that the protease inhibitor aprotinin (but not the protease inhibitor SERPINA1/alpha-1 antitrypsin) inhibited SARS-CoV-2 replication in therapeutically achievable concentrations. An analysis of proteomics and translatome data indicated that SARS-CoV-2 replication is associated with a downregulation of host cell protease inhibitors. Hence, aprotinin may compensate for downregulated host cell proteases during later virus replication cycles. Aprotinin displayed anti-SARS-CoV-2 activity in different cell types (Caco2, Calu-3, and primary bronchial epithelial cell air–liquid interface cultures) and against four virus isolates. In conclusion, therapeutic aprotinin concentrations exert anti-SARS-CoV-2 activity. An approved aprotinin aerosol may have potential for the early local control of SARS-CoV-2 replication and the prevention of COVID-19 progression to a severe, systemic disease.

## 1. Introduction

Severe acute respiratory syndrome virus 2 (SARS-CoV-2), a novel betacoronavirus, causes a respiratory disease and pneumonia called coronavirus disease 19 (COVID-19) and is the cause of a current pandemic responsible for millions of cases and hundreds of thousands of deaths [1,2,3,4,5,6,7]. Drugs for the treatment of COVID-19 are urgently needed.

Cell entry of coronaviruses is mediated by the interaction of the viral spike (S) protein with their host cell receptors, which differ between different coronaviruses [8]. For example, Middle East respiratory syndrome coronavirus (MERS-CoV) uses dipeptidyl peptidase 4 (DPP4) as a cellular receptor [8]. Host cell entry of SARS-CoV-2 and of the closely related severe acute respiratory syndrome virus (SARS-CoV) is mediated by angiotensin-converting enzyme 2 (ACE2) [8,9,10]. S binding to ACE2 depends on S cleavage at three sites (S1, S2, and S2’) by host cell proteases, typically by the transmembrane serine protease 2 (TMPRSS2), and can be inhibited by serine protease inhibitors [9,10]. Camostat was the first serine protease inhibitor that was shown to inhibit TMPRSS2 [9]. Subsequently, additional TMPRSS2 inhibitors, including nafamostat and Arbidol derivatives, were demonstrated to interfere with SARS-CoV-2 internalization into host cells [11,12,13].

Aprotinin is a serine protease inhibitor, which has previously been shown to inhibit TMPRSS2 and has been suggested as a treatment option for influenza viruses and coronaviruses [14,15]. Herein, we investigated the effects of aprotinin against SARS-CoV-2.

## 2. Materials and Methods

### 2.1. Drugs

SERPINA1/alpha-1 antitrypsin (Prolastin) was obtained from Grifols (Barcelona, Spain). Aprotinin was purchased from Sigma-Aldrich (Darmstadt, Germany)).

### 2.2. Cell Culture

The Caco2 cell line was obtained from DSMZ (Braunschweig, Germany), and Calu-3 from ATCC (Manassas, VA, US). The cells were grown at 37 °C in minimal essential medium (MEM) supplemented with 10% fetal bovine serum (FBS), 100 IU/mL of penicillin, and 100 μg/mL of streptomycin. All culture reagents were purchased from Sigma-Aldrich. Cells were regularly authenticated by short tandem repeat (STR) analysis and tested for mycoplasma contamination.

Lung tissue for the isolation of primary epithelial cells was provided by the Hannover Medical School, Institute of Pathology (Hannover, Germany). The use of tissue was approved by the ethics committee of the Hannover Medical School (MHH, Hannover, Germany, number 2701–2015) and was in compliance with The Code of Ethics of the World Medical Association. Primary bronchial epithelial cells were isolated from the lung explant tissue of a patient with lung emphysema as described previously [16]. All patients or their next of kin gave written informed consent for the use of their lung tissue for research. Basal cells were expanded in Keratinocyte-SFM medium supplemented with bovine pituitary extract (25 µg/mL), human recombinant epidermal growth factor (0.2 ng/mL, all from Gibco, Schwerte, Germany), isoproterenol (1 nM, Sigma), Antibiotic/Antimycotic Solution (Sigma-Aldrich), and MycoZap Plus PR (Lonza, Cologne, Germany) and cryopreserved until further use.

For differentiation, the cells were thawed and passaged once in PneumaCult-Ex Medium (StemCell Technologies, Cologne, Germany) and then seeded on transwell inserts (12-well plate, Sarstedt, Nümbrecht, Germany) at 4 × 10^4^ cells/insert. Once the cell layers reached confluency, the medium on the apical side of the transwell was removed, and medium in the basal chamber was replaced with PneumaCult ALI Maintenance Medium (StemCell Technologies), including Antibiotic/Antimycotic Solution (Sigma-Aldrich) and MycoZap Plus PR (Lonza). During a period of four weeks, the medium was changed and the cell layers were washed with PBS every other day. Criteria for successful differentiation were the development of ciliated cells and ciliary movement, an increase in transepithelial electric resistance indicative of the formation of tight junctions, and mucus production.

### 2.3. Virus Infection

The isolates SARS-CoV-2/1/Human/2020/Frankfurt (SARS-CoV-2/FFM1), SARS-CoV-2/2/Human/2020/Frankfurt (SARS-CoV-2/FFM2), SARS-CoV-2/6/Human/2020/Frankfurt (SARS-CoV-2/FFM6), and SARS-CoV-2/7/Human/2020/Frankfurt (SARS-CoV-2/FFM7) were isolated and cultivated in Caco2 cells as previously described [17,18]. Virus titers were determined as TCID50/mL in confluent cells in 96-well microtiter plates [19,20].

### 2.4. Antiviral Assay

Confluent cell cultures were infected with SARS-CoV-2 in 96-well plates at a multiplicity of infection (MOI) of 0.01 in the absence or presence of the drug. The cytopathogenic effect (CPE) was assessed visually 48 h post-infection [19]. Concentrations that inhibited CPE formation by 50% (IC_50_) were determined using CalcuSyn (Biosoft, Cambridge, UK).

### 2.5. Viability Assay

Cell viability was determined by 3-(4,5-dimethylthiazol-2-yl)-2,5-diphenyltetrazolium bromide (MTT) assay modified after Mosman [21], as previously described [22]. Confluent cell cultures in 96-well plates were incubated with the drug for 48 h. Then, 25 µL of MTT solution (2 mg/mL (*w*/*v*) in PBS) were added per well, and the plates were incubated at 37 °C for an additional 4 h. After this, the cells were lysed using 200 µL of a buffer containing 20% (*w*/*v*) sodium dodecylsulfate and 50% (*v*/*v*) *N*,*N*-dimethylformamide with the pH adjusted to 4.7 at 37 °C for 4 h. Absorbance was determined at 570 nm for each well using a 96-well multiscanner (Tecan, Crailsheim, Germany). After subtracting of the background absorption, the results are expressed as percentage viability relative to control cultures that received no drug. Drug concentrations that inhibited cell viability by 50% (CC_50_) were determined using CalcuSyn (Biosoft).

### 2.6. Immunostaining for SARS-CoV-2 S Protein

Immunostaining was performed as previously described [23], using a monoclonal antibody directed against SARS-CoV-2 S protein (1:1500 dilution, Sino Biological, Eschborn, Germany) 24 h post-infection.

### 2.7. Caspase 3/7 Activation

Caspase 3/7 activation was determined using the Caspase-Glo^®^ 3/7 Assay (Promega, Walldorf, Germany) according to the manufacturer’s instructions.

### 2.8. qPCR

SARS-CoV-2 RNA from the cell culture supernatant samples was isolated using AVL buffer and the QIAamp Viral RNA Kit (Qiagen, Hilden, Germany) according to the manufacturer’s instructions. Absorbance-based quantification of the RNA yield was performed using the Genesys 10S UV-Vis Spectrophotometer (Thermo Fisher Scientific, Dreieich, Germany). RNA was subjected to OneStep qRT-PCR analysis using the Luna Universal One-Step RT-qPCR Kit (New England Biolabs, Frankfurt am Main, Germany) and a CFX96 Real-Time System, C1000 Touch Thermal Cycler (Bio-Rad, Feldkirchen, Germany). Primers were adapted from the WHO protocol29 targeting the open reading frame for RNA-dependent RNA polymerase (RdRp): RdRP_SARSr-F2 (GTG ARA TGG TCA TGT GTG GCG G) and RdRP_SARSr-R1 (CAR ATG TTA AAS ACA CTA TTA GCA TA) using 0.4 µM per reaction. Standard curves were created using plasmid DNA (pEX-A128-RdRP) harboring the corresponding amplicon regions for RdRp target sequence according to GenBank Accession number NC_045512. For each condition, three biological replicates were used. The mean and standard deviation were calculated for each group.

### 2.9. Western Blot

Cells were lysed using Triton-X-100 sample buffer (Sigma-Aldrich), and proteins were separated by SDS-PAGE. Detection occurred by using specific antibodies against SARS-CoV-2 N (1:1000 dilution, SARS-CoV-2 Nucleocapsid Antibody, Rabbit monoclonal antibody (Mab), #40143-R019, Sino Biological), ACE2 (1:500 dilution, Anti-ACE2 antibody, #ab15348, Abcam, Berlin, Germany), TMPRSS2 (1:1000 dilution, Recombinant Anti-TMPRSS2 antibody [EPR3861], #ab92323, Abcam), and GAPDH (1:1000 dilution, Anti-G3PDH Human Polyclonal Antibody, #2275-PC-100, Trevigen, Wiesbaden, Germany). Protein bands were visualized by laser-induced fluorescence using an infrared scanner for protein quantification (Odyssey, Li-Cor Biosciences, Bad Homburg, Germany).

### 2.10. Sample Preparation for LC–MS

Preparation of samples was performed as previously described [24] and labeled with TMTpro multiplexing reagents.

### 2.11. Targeted Analysis by SPS–MS^3^

Mass spectrometry data were acquired in centroid mode on an Orbitrap Fusion Lumos mass spectrometer hyphenated to an easy-nLC 1200 nano HPLC system using a nanoFlex ion source (ThermoFisher Scientific) applying a spray voltage of 2.6 kV with the transfer tube heated to 300 °C and a funnel RF of 30%. Internal mass calibration was enabled (lock mass 445.12003 *m*/*z*). Peptides were separated on a self-made, 32 cm long, 75 µm ID fused-silica column, packed in house with 1.9 µm C18 particles (ReproSil-Pur, Dr. Maisch, Ammerbuch-Entringen, Germany) and heated to 50 °C using an integrated column oven (Sonation, Biberach, Germany). The HPLC solvents consisted of 0.1% formic acid in water (Buffer A) and 0.1% formic acid with 80% acetonitrile in water (Buffer B).

Dependent scans were performed on precursors matching a mass list of viral peptides modified with TMTpro reagents and their charge states (mass tolerance was set to 5 ppm for matching precursors). Peptides were eluted by a non-linear gradient from 5% to 40% B over 30 min, followed by a step-wise increase to 95% B in 6 min, which was held for another 9 min. Full scan MS spectra (350–1500 *m/z*) were acquired with a resolution of 120,000 at *m*/*z* 200, a maximum injection time of 100 ms, and an automatic gain control (AGC) target value of 4 × 10^5^. The 10 most intense precursors matching the target list per full scan were selected for fragmentation (“Top 10”) and isolated with a quadrupole isolation window of 0.4 Th. MS2 scans were performed in the Orbitrap using a maximum injection time of 300 ms, an AGC target value of 1.5 × 10^4^, and fragmented using HCD with a normalized collision energy (NCE) of 35% and a fixed first mass of 110 *m*/*z*. Repeated sequencing of already acquired precursors was limited by setting a dynamic exclusion of 20 s and 10 ppm and advanced peak determination was deactivated.

### 2.12. Data Analysis

RAW data was processed with Proteome Discoverer 2.4 software. HCD-fragmented spectra were searched against a SARS-CoV-2 proteome FASTA file (UniProt pre-realease) by SequestHT and the false discovery rate (FDR) was calculated using a target/decoy-based approach. TMTpro reporter abundances were extracted and used for plotting and statistical analysis.

### 2.13. Data Availability

The mass spectrometry proteomics data were deposited to the ProteomeXchange Consortium via the PRIDE [25] partner repository with the dataset identifier PXD019950.

## 3. Results

### 3.1. The Protease Inhibitor Aprotinin Exerts Superior Anti-SARS-CoV-2 Activity Relative to the Endogenous Protease Inhibitor SERPINA1/alpha-1 Antitrypsin

We compared the anti-SARS-CoV-2 activity of aprotinin [15,26] and SERPINA1/alpha-1 antitrypsin, an endogenous protease inhibitor that is available as a pharmaceutical preparation for the treatment of alpha-1 antitrypsin deficiency [27], against three different SARS-CoV-2 isolates from two lineages (L: SARS-CoV-2/FFM1 and SARS-CoV-2/FFM2; GR: SARS-CoV-2/FFM6) [18]. SARS-CoV-2/FFM1 and SARS-CoV-2/FFM2 were isolated from patients in Hubei province in China, while SARS-CoV/FFM6 was derived from an Italian patient [18].

The aprotinin concentrations that inhibited the formation of cytopathogenic effects (CPEs) by 50% (IC_50_) in SARS-CoV-2-infected Caco2 cells ranged from 0.81 µM (SARS-CoV-2/FFM2) to 1.03 µM (SARS-CoV-2/FFM1) across the three tested SARS-CoV-2 isolates, whereas SERPINA1/alpha-1 antitrypsin did not show significant antiviral effects in the tested concentrations up to 20 µM (Figure 1A). Similar effects were observed by cell staining for SARS-CoV-2 S protein (Figure 1B and Appendix A, Table 1). Quantification of genomic SARS-CoV-2 RNA using qPCR confirmed that aprotinin inhibits SARS-CoV-2 replication (Figure 1C). Aprotinin (20 µM) reduced the genomic RNA levels of SARS-CoV-2/FFM1 by 900-fold, those of SARS-CoV-2/FFM2 by 237-fold, and those of SARS-CoV-2/FFM6 by 584-fold.

Both aprotinin and SERPINA1/alpha-1 antitrypsin are trypsin inhibitors [26,28]. To verify the integrity of the used protease inhibitor samples, we tested their capacity to antagonize trypsin and enable Caco2 and A549 cell adhesion. The results confirmed that both protease inhibitors are active (Appendix A). Taken together, these findings indicate differences in the protease inhibitor spectrum of aprotinin and SERPINA1/alpha-1 antitrypsin that result in different effects on SARS-CoV-2 replication.

### 3.2. Quantification of the Antiviral Effects of Aprotinin by Measuring SARS-CoV-2-Induced Caspase 3/7 Activation

Different viruses, including SARS-CoV-2, have been shown to induce caspase 3 activation [29,30,31,32], and virus-induced caspase 3 activation has been used as read-out in assays that quantify the antiviral effects of drug candidates [31]. Hence, we used the Caspase-Glo^®^ 3/7 Assay (Promega) as an additional quantitative method to determine the anti-SARS-CoV-2 activity of aprotinin. The results confirmed those obtained by CPE formation and S expression resulting in similar IC_50_ values (Figure 2, Table 1).

### 3.3. Aprotinin Inhibits Virus Entry

Protease inhibitors were suggested to interfere with SARS-CoV-2 replication predominantly as entry inhibitors that prevent S cleavage and activation [15]. In agreement, aprotinin addition after a one-hour adsorption period did not significantly interfere with SARS-CoV-2 replication in one round of a replication assay, in which virus titers were determined 8 h post-infection with an MOI of 0.1 (Figure 3A). In contrast, remdesivir, which was anticipated to interfere with the replication of the viral genome, inhibited SARS-CoV-2 replication when added post-infection (Figure 3A).

### 3.4. Aprotinin May Interfere with SARS-CoV-2-Mediated Downregulation of Host Cell Protease Inhibitors

Notably, aprotinin exerted similar anti-SARS-CoV-2 effects when added before or after infection of Caco2 cells with a lower MOI (0.01) in a 48 h assay (Figure 3B). In this format, aprotinin probably inhibits the later rounds of SARS-CoV-2 replication, but other mechanisms may also contribute.

Host cell protease inhibitors interfere with the activity of proteases such as TMPRSS2 [33,34] that mediate SARS-CoV-2 cell entry by cleaving and activating the viral S protein [9,11,12]. An analysis of the effect of SARS-CoV-2 infection on host cell protease inhibitors using proteomics data from SARS-CoV-2-infected Caco2 cells [35] showed that the endogenous protease inhibitors SPINT1 (Kunitz-type protease inhibitor 1), SPINT2 (Kunitz-type protease inhibitor 2), and SERPINA1 (alpha-1-antitrypsin) are present at lower levels in SARS-CoV-2-infected cells than in non-infected control cells 24 h post-infection (Figure 4A). Translatome data from the same dataset [35] indicated that the translation of SERPINA1 and SPINT2 (but not that of SPINT1) is also reduced in SARS-CoV-2-infected cells (Figure 4B). Hence, SARS-CoV-2 infection results in the downregulation of endogenous protease inhibitors, which may support SARS-CoV-2 replication. Thus, compensation for downregulated endogenous protease inhibitors may contribute to the antiviral effects of aprotinin.

### 3.5. Aprotinin Exerts Anti-SARS-CoV-2 Activity in Air–Liquid Interface (ALI) Cultures from Primary Bronchial Epithelial Cells

We also investigated the effects of aprotinin in SARS-CoV-2-infected air–liquid interface (ALI) cultures from primary bronchial epithelial cells. A targeted proteomics assay demonstrated that aprotinin 20 µM suppressed the expression of the SARS-CoV-2 proteins N (nucleocapsid protein) and M (membrane protein) in SARS-CoV-2-infected ALI cultures (Figure 5A, Appendix A). The results for N were confirmed by Western blots in the ALI cultures infected with SARS-CoV-2/FFM7 (Figure 5B). SARS-CoV-2/FFM7 (G lineage) is an alternative isolate derived from a patient from Israel [18]. Aprotinin also suppressed SARS-CoV-2 S expression in SARS-CoV-2/FFM7-infected Calu-3 lung adenocarcinoma cells (Appendix A).

## 4. Discussion

Herein, we showed that aprotinin inhibits SARS-CoV-2 replication predominantly as an entry inhibitor, probably via interfering with SARS-CoV-2 S activation by TMPRSS2. Notably, SERPINA1/alpha-1 antitrypsin, which is available as a pharmaceutical preparation for the treatment of alpha-1 antitrypsin deficiency [27], did not inhibit SARS-CoV-2 replication in the same concentration range. Further investigations will have to elucidate the differences between aprotinin and SERPINA1/alpha-1 antitrypsin that are responsible for the discrepancy in anti-SARS-CoV-2 activity. Notably, SERPINA1/alpha-1 antitrypsin was shown to inhibit TMPRSS2 in an enzymatic assay and is suggested as an antiviral treatment for COVID-19 [36]. A clinical trial testing SERPINA1/alpha-1 antitrypsin for the treatment of COVID-19 has recently been started (ClinicalTrials.gov Identifier: NCT04385836). Based on our data, however, SERPINA1/alpha-1 antitrypsin is not expected to exert direct antiviral effects in COVID-19 patients. Our findings also indicate that antiviral therapy candidates should be tested for their effects on complete replication-competent viruses in permissive cells.

Aprotinin exerted anti-SARS-CoV-2 effects in three cell culture models (Caco2, Calu-3, and air–liquid interface cultures from primary bronchial epithelial cells) and against three SARS-CoV-2 strains (FFM1, FFM2, FFM6, and FFM7). Notably, another study became available during the revision of our manuscript that detected anti-SARS-CoV-2 activity of aprotinin in Calu-3 cells [37]. Our findings are also in agreement with studies that reported other TMPRSS2 inhibitors to inhibit SARS-CoV-2 entry and replication [11,12,13]. In addition, furin has been shown to cleave and activate SARS-CoV-2 S and furin inhibitors have been demonstrated to exert anti-SARS-CoV-2 effects [38].

Endogenous protease inhibitors may interfere with the activation of virus surface proteins such as S by host cell proteases [9,11,12,33,34]. Our analysis of proteomics and translatome data from SARS-CoV-2-infected Caco2 cells [35] revealed a downregulation of endogenous protease inhibitors in response to SARS-CoV-2 infection, which may contribute to efficient SARS-CoV-2 replication. In addition to entry inhibition, compensation for downregulated endogenous proteases may, hence, further contribute to the antiviral activity of aprotinin during later rounds of SARS-CoV-2 replication.

The clinical potency of aprotinin is typically measured in kallikrein inhibitor units (KIUs) [14,39]. Therapeutic aprotinin plasma levels were described to reach 147 ± 61 KIU/mL after the administration of 1,000,000 KIU [39]. Moreover, an aerosol preparation of aprotinin, which is likely to result in increased local aprotinin concentrations in the lung, is approved for the treatment of influenza in Russia [14]. The aprotinin IC_50_ values for SARS-CoV-2-induced CPE formation, S expression, and apoptosis induction ranged from 0.32 to 1.65 µM, which is equivalent to 4.0 KIU and 20.6 KIU, respectively. Hence, aprotinin interferes with SARS-CoV-2 infection in therapeutically achievable concentrations.

Aprotinin exerts pro- and antithrombotic effects by balancing fibrinolysis and thrombus formation and is approved for the prevention of blood loss during surgery. It interferes with the fibrinolysis of established thrombi by plasmin, but also inhibits contact-activated thrombus formation in the blood stream [26,40,41,42,43]. Late-stage, severe COVID-19 disease has been associated with disseminated intravascular coagulation and thrombosis (COVID-19-related coagulopathy) [44]. Based on the available data, it is not clear whether aprotinin may exert pro- or antithrombotic effects in patients suffering from COVID-19-related coagulopathy. Thus, aprotinin would have to be considered with care for such patients. 

However, antiviral treatment may anyway be of limited impact in late-stage COVID-19 disease, during which, damage is anticipated to be largely caused by immunopathology and not by virus replication [42,45,46]. Hence, the main potential of antiviral drugs may lie in the early treatment of COVID-19 patients to suppress virus replication and, through this, to prevent COVID-19 progression into a severe, life-threatening disease. Local aprotinin therapy of the airways and the lungs using an aerosol, which is clinically approved in Russia and has been reported to be very well tolerated in influenza patients [14], may have particular potential as such an antiviral treatment for early stage COVID-19 disease. Notably, aprotinin may additionally prevent the very early stages of lung injury by inhibition of matrix metalloproteinases and, in turn, of the cytokine storm that eventually results in severe, systemic COVID-19 disease [26].

## 5. Conclusions

In conclusion, therapeutic aprotinin concentrations inhibit SARS-CoV-2 replication as entry inhibitors and by compensating for downregulated cellular protease inhibitors during later replication cycles. Local treatment of the respiratory tract using an aprotinin aerosol, which is approved in Russia for the treatment of influenza [14], may be a particularly promising strategy to suppress virus replication and lung injury early and to prevent COVID-19 progression into a severe, systemic disease.

## Figures and Tables

**Figure 1 cells-09-02377-f001:**
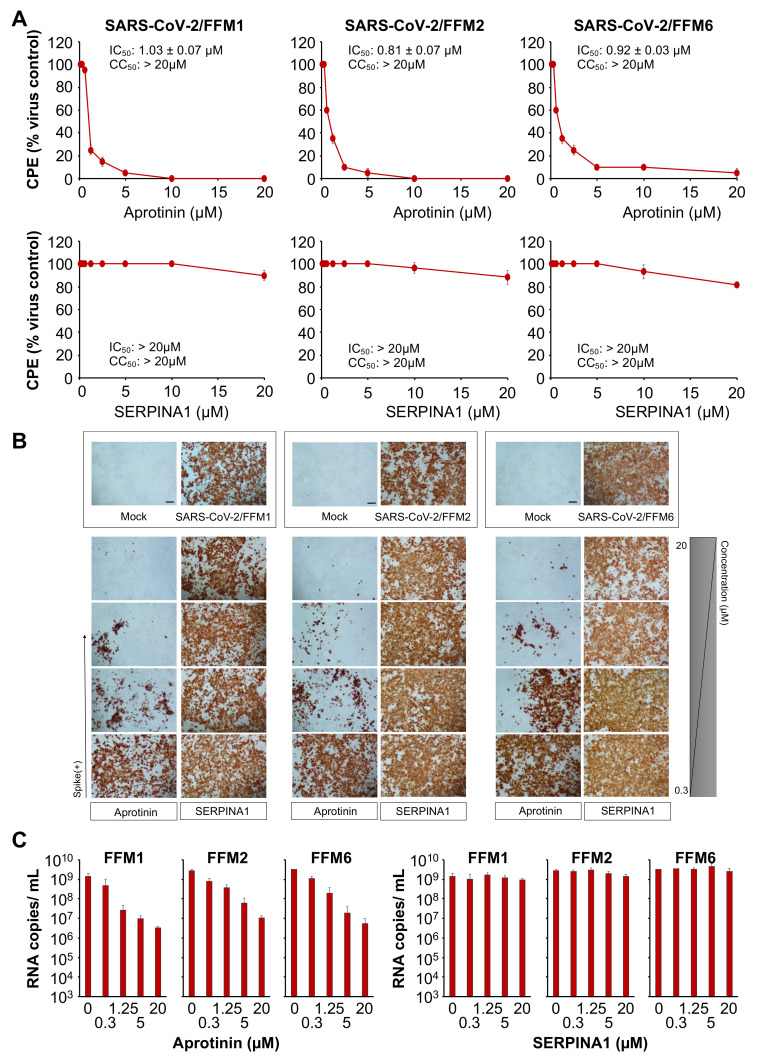
Anti-severe acute respiratory syndrome virus 2 (SARS-CoV-2) effects of aprotinin and SERPINA1/alpha-1 antitrypsin. (**A**) Concentration-dependent effects of aprotinin and SERPINA1/alpha-1 antitrypsin on SARS-CoV-2-induced cytopathogenic effect (CPE) formation determined 48 h post-infection in Caco2 cells infected at a multiplicity of infection (MOI) of 0.01 with the three different SARS-CoV-2 isolates. The viability of the Caco2 cells was 84.3 ± 2.7% relative to the untreated control in the presence of 20 µM of aprotinin. (**B**) Immunostaining for the SARS-CoV-2 S protein in aprotinin- and SERPINA1/alpha-1 antitrypsin-treated Caco2 cells infected at an MOI of 0.01 with the three different SARS-CoV-2 isolates as determined 48 h post-infection. The protease inhibitors were tested at four concentrations in 1:4 dilution steps ranging from 20 to 0.3125 µM. A quantification is provided in Appendix A. (**C**) Copy numbers of genomic RNA in Caco2 cells infected with different SARS-CoV-2 isolates (MOI of 0.01) in response to treatment with aprotinin or SERPINA1/alpha-1 antitrypsin as determined 48 h post-infection. FFM1, 1/Human/2020/Frankfurt; FFM2, 2/Human/2020/Frankfurt; FFM6, 6/Human/2020/Frankfurt.

**Figure 2 cells-09-02377-f002:**
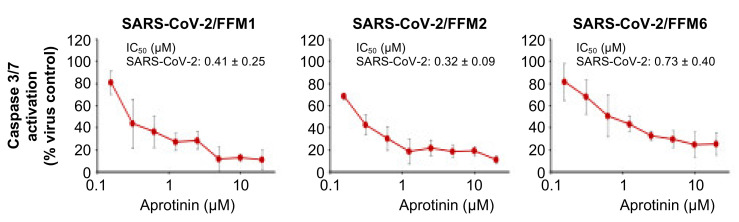
Effects of aprotinin on SARS-CoV-2-induced caspase 3/7 activation. Caspase 3/7 activity was determined in Caco2 cells infected with different SARS-CoV-2 isolates (MOI of 0.01) 48 h post-infection.

**Figure 3 cells-09-02377-f003:**
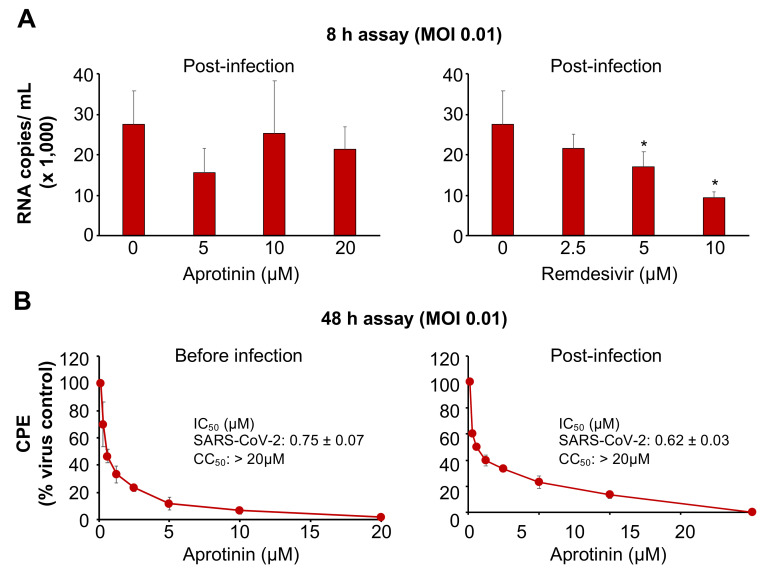
Anti-SARS-CoV-2 effects of aprotinin when administered post-infection. For post-infection experiments, the cells were incubated with the virus for a one-hour adsorption period. Then, the cells were washed three times in PBS prior to the addition of the drug. (**A**) The effects of aprotinin and the RNA polymerase inhibitor remdesivir (a positive control drug that interferes with virus replication after virus entry) on virus replication as determined by qPCR in SARS-CoV-2/FFM1 (MOI of 0.1)-infected Caco2 cells 8 h post-infection (after approximately one round of replication). * *p* < 0.05 as determined by one-way ANOVA and Dunnett’s multiple comparison test. (**B**) The effects of aprotinin on cytopathogenic effect (CPE) formation in SARS-CoV-2/FFM1 (MOI of 0.01)-infected Caco2 cells were determined 48 h post-infection.

**Figure 4 cells-09-02377-f004:**
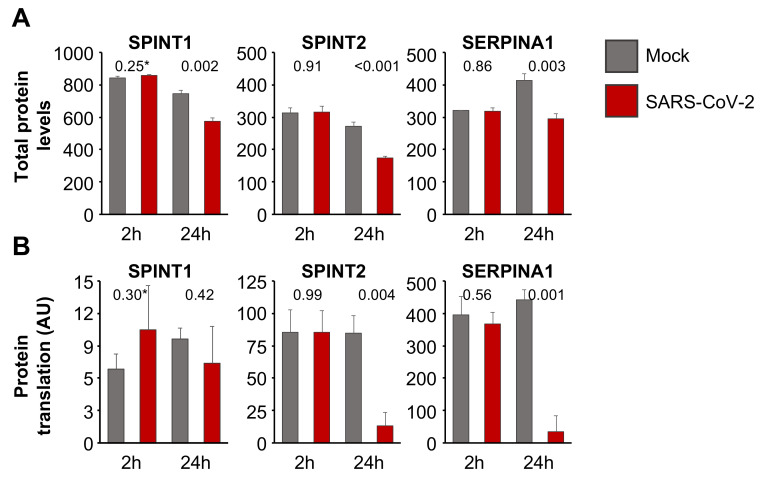
Regulation of host cell protease inhibitors in SARS-CoV-2-infected cells. (**A**) Total protein levels based on a publicly available proteomics dataset [35], indicating cellular levels of endogenous protease inhibitors in SARS-CoV-2 (MOI of 1)-infected Caco2 cells 2 h and 24 h post-infection. Data were normalized using summed intensity normalization for sample loading, followed by internal reference scaling and trimmed mean of M normalization. * *p*-values as determined using a two-sided Student’s *t*-test. (**B**) Mean protein translation of endogenous protease inhibitors in arbitrary units (AU) (normalized and corrected summed peptide spectrum matches (PSMs) were averaged) in SARS-CoV-2 (MOI of 1)-infected Caco2 cells 2 h and 24 h post-infection based on a publicly available translatome dataset [35]. * *p*-values as determined using a two-sided Student’s *t*-test.

**Figure 5 cells-09-02377-f005:**
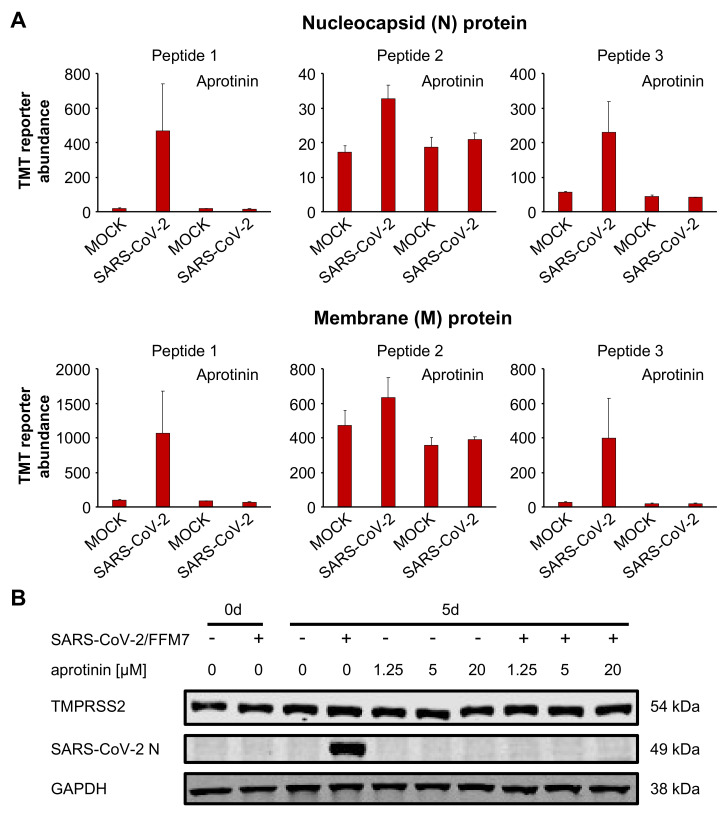
Antiviral effects of aprotinin in SARS-CoV-2-infected air–liquid interface (ALI) cultures from primary bronchial epithelial cells. (**A**) Abundance of the SARS-CoV-2 proteins N (nucleocapsid) and M (membrane) in primary bronchial epithelial cell ALI cultures infected with SARS-CoV-2/FFM1 (MOI of 1) in the presence or absence of aprotinin (20 µM) as determined 5 days post-infection by multiplexed mass spectrometry analysis using acquisition targeting of previously identified viral peptides modified with TMTpro. The detailed data are presented in Appendix A. (**B**) Western blots indicating cellular SARS-CoV-2 N and TMPRSS2 levels in primary bronchial epithelial cell ALI cultures infected with SARS-CoV-2/7/Human/2020/Frankfurt (FFM7) (MOI of 1) in the presence or absence of aprotinin as detected 5 days post infection. GAPDH was served as the loading control. Uncropped Western blots are shown in Appendix A.

**Table 1 cells-09-02377-t001:** Aprotinin concentrations that reduce SARS-CoV-2-induced cytopathogenic effect (CPE) formation, SARS-CoV-2 spike (S) levels, and SARS-CoV-2-induced caspase 3/7 activation by 50% (IC_50_) as determined in Caco2 cells infected with different SARS-CoV-2 isolates (MOI of 0.01) 48 h post-infection.

	IC_50_ (µM)		
	FFM1	FFM2	FFM6
CPE formation	1.03 ± 0.07	0.81 ± 0.07	0.92 ± 0.03
S levels	0.79 ± 0.15	1.04 ± 0.21	1.65 ± 0.30
Caspase 3/7 activation	0.41 ± 0.25	0.32 ± 0.09	0.73 ± 0.40

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
