# Peer review of "Aprotinin Inhibits SARS-CoV-2 Replication"

_cells, 2020, doi:10.3390/cells9112377_

Round 1

Reviewer 1 Report

This study, titled “Aprotinin inhibits SARS-CoV-2 replication by compensating for the virus-induced downregulation of host cell protease inhibitors,” by Bojkova et al is of important research during the current COVID-19 pandemic. This study explains the potential of aprotinin, a protease inhibitor, to provide protection against SARS-CoV-2 infection. It also shows that this effect is a least in part specific as anti-trypsin did not show the same effect. 

This study demonstrates potential, but should modified to further improve the conclusions made as well as the overall readability:

  • For Figure 2A, the viability data could be included. Also, without achieving a concentration that is cytotoxic it is not possible to determine an SI, is the next increase in aprotonin going to be cytotoxic, or is much higher concentration possible. This is important, despite the activity occurring at a reasonable and achievable concentration.
  • Possibly a log format on the X-axis would separate the close points from 0 to 5 uM, it is unclear if this would improve or detract from the presentation of the data. 

Minor suggestions:

  • There are some grammatical errors throughout the paper that should be corrected. Some of these errors include past/present inconsistency (ex. Line 35-35 should read “Protease inhibitors have been regarded as virus entry inhibitors, which prevents the cleavage…”), split infinitives (ex. Line 46 “possible also”) and missing articles (ex. Line 83 should read “For differentiation, the cells…; Line 90 should read “an increase in transepithelial”). Please correct these throughout the text.
  • Define all abbreviated terms before using them (ex. Line 88 “PBS”, Line 99 “MOI.”).
  • In Section 2.5 Viability assay, Line 111, IC50 should be defined as “percent CPE inhibited by 50%,” as this is how you describe your results in Line 193, 210, etc. IC50 description should also be moved to Section 2.4 Antiviral assay.
  • In Section 2.5, CC50 was not defined. Please define it.
  • Add Materials section for the Statistical Analysis that was performed throughout the study. And define the statistical significances of the p values.
  • Be consistent on how you define “SERPINA1/ alpha-1 antitrypsin” (Compare line 164 to 185).
  • For Section 3.2, Line 196 where you state “SERPINA1/ alpha-1 antitrypsin only started to show minor antiviral effects at 20uM”. When visually looking at Figure 2A, and taking into consideration the error bars, it appears there was no antiviral effects at all at 20uM. Please change the wording to reflect this.
  • For Figure 3, the resolution appears to be low. 
  • In Line 314, you mention ‘therapeutically achievable concentrations.’ Please elaborate on what these concentrations are, and how aprotinin achieves these levels.

Author Response

Introductory note by the authors:

This is an unusual revision, since reviewer 2 asked us to perform additional time-of-addition experiments with a read-out after eight hours, a time point that roughly reflects one round of replication. Unexpectedly for us, these experiments revealed that entry inhibition makes a much more pronounced contribution to the anti-SARS-CoV-2 effects of aprotinin than we had originally anticipated. Hence, we had to revise our manuscript to an extent that makes it impossible to highlight the changes in the manuscript in a reasonable manner. However, we added a substantial amount of extra data and feel that this has substantially improved the manuscript. Please find the point-by-point responses to your requests below.

Reviewer comments:

This study, titled “Aprotinin inhibits SARS-CoV-2 replication by compensating for the virus-induced downregulation of host cell protease inhibitors,” by Bojkova et al is of important research during the current COVID-19 pandemic. This study explains the potential of aprotinin, a protease inhibitor, to provide protection against SARS-CoV-2 infection. It also shows that this effect is a least in part specific as anti-trypsin did not show the same effect.

This study demonstrates potential, but should modified to further improve the conclusions made as well as the overall readability:

For Figure 2A, the viability data could be included. Also, without achieving a concentration that is cytotoxic it is not possible to determine an SI, is the next increase in aprotonin going to be cytotoxic, or is much higher concentration possible. This is important, despite the activity occurring at a reasonable and achievable concentration.

Possibly a log format on the X-axis would separate the close points from 0 to 5 uM, it is unclear if this would improve or detract from the presentation of the data.

Authors' response:

This was done. The following sentence was added to the figure legend (now Figure 1A, lines 194/195):

"The viability of Caco2 cells was 84.3 ± 2.7% relative to untreated control in the presence of aprotinin 20µM."

Minor suggestions:

There are some grammatical errors throughout the paper that should be corrected. Some of these errors include past/present inconsistency (ex. Line 35-35 should read “Protease inhibitors have been regarded as virus entry inhibitors, which prevents the cleavage…”), split infinitives (ex. Line 46 “possible also”) and missing articles (ex. Line 83 should read “For differentiation, the cells…; Line 90 should read “an increase in transepithelial”). Please correct these throughout the text.

Define all abbreviated terms before using them (ex. Line 88 “PBS”, Line 99 “MOI.”).

In Section 2.5 Viability assay, Line 111, IC50 should be defined as “percent CPE inhibited by 50%,” as this is how you describe your results in Line 193, 210, etc. IC50 description should also be moved to Section 2.4 Antiviral assay.

In Section 2.5, CC50 was not defined. Please define it.

Add Materials section for the Statistical Analysis that was performed throughout the study. And define the statistical significances of the p values.

Be consistent on how you define “SERPINA1/ alpha-1 antitrypsin” (Compare line 164 to 185).

Authors response:

We have corrected these mistakes.

For Section 3.2, Line 196 where you state “SERPINA1/ alpha-1 antitrypsin only started to show minor antiviral effects at 20uM”. When visually looking at Figure 2A, and taking into consideration the error bars, it appears there was no antiviral effects at all at 20uM. Please change the wording to reflect this.

Authors response:

The wording was changed as follows (lines 177/178):

"whereas SERPINA1/ alpha-1 antitrypsin did not show significant antiviral effects in the tested concentrations up to 20µM"

For Figure 3, the resolution appears to be low.

Authors response:

We seem to have some issues with some our figures and the template. Sharp figures are provided in the supplements.

In Line 314, you mention ‘therapeutically achievable concentrations.’ Please elaborate on what these concentrations are, and how aprotinin achieves these levels.

Authors response:

There is no a paragraph (lines 300-307) explaining this:

"The clinical potency of aprotinin is typically measured in kallikrein inhibitor units (KIU) [12,38]. Therapeutic aprotinin plasma levels were described to reach 147 ± 61KIU/mL after the administration of 1,000,000KIU [38]. Moreover, an aerosol preparation of aprotinin, which is likely to result in increased local aprotinin concentrations in the lung, is approved for the treatment of influenza in Russia [12]. The aprotinin IC50 values for SARS-CoV-2-induced CPE formation, S expression, and apoptosis induction ranged from 0.32µM to 1.65µM, which is equivalent to 4.0KIU and 20.6KIU, respectively. Hence, aprotinin interferes with SARS-CoV-2 infection in therapeutically achievable concentrations."

Reviewer 2 Report

In the article “Aprotinin inhibits SARS-CoV-2 replication by compensating for the virus-induced down regulation of host cell protease inhibitors” by Bojkova et al, the authors evaluate the use of protease inhibitors for the treatment of COVID-19. In this work they use their previously published data to find that cellular proteases are downregulated following SARS-CoV-2 infection. Using this information, they use two available protease inhibitors to show that one (aprotinin) is able to reduce SARS-CoV-2 infection. While the idea of the story is interesting, the actual data presented seem to be very preliminary and lacking in depth.

Major comments:

  1. The paper relies on the concept that inhibition of proteases blocks viral replication however the assay used to make this statement does not allow for the conclusion. The authors use a CPE assay (Figure 3) which relies on the virus being released from cells and ENTERING neighboring cells in order to form plaques. Therefore, using this assay it is impossible to separate entry from replication. Given the results of Figure 1 and 2, they are blocking entry and as such blocking CPE. To be able to conclude on whether or not proteases impact viral replication, the authors need to perform the assay within one round of virus infection (ie 8h). This should be evaluated by both immunofluorescence and qPCR. As the authors use both of these assays in Figure 2, they are available and should be easy to include.

  1. Additionally, to show that aprotinin is affecting replication it would be necessary to show this on the protein level. This could be done by Western blot or as the lab has a history of using mass-spec, a time resolved abundance mass spec could be used to show that viral proteins are not being made/accumulated due to a block in replication.

  1. Figure 4 is very strange and out of context. These are different viruses and different cultures that are now evaluated by mass spec. It seems that these data were made for a different paper and just thrown in here to have a primary model. As the rest of the paper is using human colon cancer cells, it seems that it would make more sense to use primary intestinal cells here or if the paper included human lung cells (Calu-3) to make it more consistent. Additionally, it would be good to see immunofluorescence and qPCR data for these cultures as well.

  1. The title is misleading. In order to make the claims the authors need to actual have a rescue experiment. If they cannot provide this type of experiment, the title should be re-worded to reflect the data.

  1. The authors model the viral proteases with the compounds in Figure 5. This modeling is very interesting but should be followed up with mutagenesis (since there are several reverse genetics system available) to show that aprotinin is acting on the virus itself and not on a cellular target.

Minor comments:

  1. Missing qPCR information from materials and methods.

  1. In the discussion, line 298 should read assay not essay.

  1. The discussion and the conclusion have three sentences about the Russia use of this inhibitor. For a one-page discussion this statement does not need to be repeated these many times.

Author Response

Introductory note by the authors:

Many thanks for your comments that turned out to be extremely valuable and helpful. The suggested eight-hour time of addition experiment indeed showed that entry inhibition is much more important for the antiviral activity of aprotinin than we had anticipated. Hence, we had to revise our manuscript to an extent that makes it impossible to highlight the changes in a reasonable manner. However, we added a substantial amount of extra data and feel that this has substantially improved the manuscript. Please find the point-by-point responses to your requests below.

In the article “Aprotinin inhibits SARS-CoV-2 replication by compensating for the virus-induced down regulation of host cell protease inhibitors” by Bojkova et al, the authors evaluate the use of protease inhibitors for the treatment of COVID-19. In this work they use their previously published data to find that cellular proteases are downregulated following SARS-CoV-2 infection. Using this information, they use two available protease inhibitors to show that one (aprotinin) is able to reduce SARS-CoV-2 infection. While the idea of the story is interesting, the actual data presented seem to be very preliminary and lacking in depth.

Major comments:

The paper relies on the concept that inhibition of proteases blocks viral replication however the assay used to make this statement does not allow for the conclusion. The authors use a CPE assay (Figure 3) which relies on the virus being released from cells and ENTERING neighboring cells in order to form plaques. Therefore, using this assay it is impossible to separate entry from replication. Given the results of Figure 1 and 2, they are blocking entry and as such blocking CPE. To be able to conclude on whether or not proteases impact viral replication, the authors need to perform the assay within one round of virus infection (ie 8h). This should be evaluated by both immunofluorescence and qPCR. As the authors use both of these assays in Figure 2, they are available and should be easy to include.

Authors response:

The new time-of-addition experiments are presented in Figure 3A.

Additionally, to show that aprotinin is affecting replication it would be necessary to show this on the protein level. This could be done by Western blot or as the lab has a history of using mass-spec, a time resolved abundance mass spec could be used to show that viral proteins are not being made/accumulated due to a block in replication.

Authors response:

Western blot data are presented in Figure 5. We feel confident that aprotinin inhibits virus replication and have shown this by inhibition of CPE formation (Figure 1A, Figure 3B), immunostaining for S (Figure 1B, Figure S1, Figure S3), inhibition of SARS-CoV-2-induced apoptosis (Figure 2), and qPCR (Figure 1C, Figure 3A).

Figure 4 is very strange and out of context. These are different viruses and different cultures that are now evaluated by mass spec. It seems that these data were made for a different paper and just thrown in here to have a primary model. As the rest of the paper is using human colon cancer cells, it seems that it would make more sense to use primary intestinal cells here or if the paper included human lung cells (Calu-3) to make it more consistent. Additionally, it would be good to see immunofluorescence and qPCR data for these cultures as well.

Authors response:

We struggle with this comment. A range of different models and methods would normally be regarded to increase the validity of the results. The main reason for the differences in methodology are the specificities associated with ALI cultures and the time required for their cultivation, which limits through-put. Some Calu-3 data are presented in Figure S3.

The title is misleading. In order to make the claims the authors need to actual have a rescue experiment. If they cannot provide this type of experiment, the title should be re-worded to reflect the data.

Authors response:

The title was changed to "Aprotinin inhibits SARS-CoV-2 replication"  

The authors model the viral proteases with the compounds in Figure 5. This modeling is very interesting but should be followed up with mutagenesis (since there are several reverse genetics system available) to show that aprotinin is acting on the virus itself and not on a cellular target.

Authors response:

This was removed, because we agree that this is too speculative. Moreover, we are, thanks to the experiments suggested by you, convinced now that entry inhibition is a main mechanism of anti-SARS-CoV-2 action of aprotinin.

Minor comments:

Missing qPCR information from materials and methods.

Authors response:

The method was added (lines 113-125):

"SARS-CoV-2 RNA from cell culture supernatant samples was isolated using AVL buffer and the QIAamp Viral RNA Kit (Qiagen) according to the manufacturer’s instructions. Absorbance-based quantification of the RNA yield was performed using the Genesys 10S UV-Vis Spectrophotometer (Thermo Scientific). RNA was subjected to OneStep qRT-PCR analysis using the Luna Universal One-Step RT-qPCR Kit (New England Biolabs) and a CFX96 Real-Time System, C1000 Touch Thermal Cycler. Primers were adapted from the WHO protocol29 targeting the open reading frame for RNA-dependent RNA polymerase (RdRp): RdRP_SARSr-F2 (GTG ARA TGG TCA TGT GTG GCG G) and RdRP_SARSr-R1 (CAR ATG TTA AAS ACA CTA TTA GCA TA) using 0.4 µM per reaction. Standard curves were created using plasmid DNA (pEX-A128-RdRP) harbouring the corresponding amplicon regions for RdRP target sequence according to GenBank Accession number NC_045512. For each condition three biological replicates were used. Mean and standard deviation were calculated for each group."

In the discussion, line 298 should read assay not essay.

Authors response:

This was corrected.

The discussion and the conclusion have three sentences about the Russia use of this inhibitor. For a one-page discussion this statement does not need to be repeated these many times.

Authors response:

We feel that the three mentions are justified, because they are made in different contexts.

Lines 302-304 refer to clinically achievable concentrations.

Lines 320-323 refer to the safety of the approved preparation.

Lines 329-331 is part of the conclusion. In the light of an ongoing pandemic, clinical preparations that can be repurposed are of interest.

Reviewer 3 Report

The manuscript shows that the serine protease inhibitor aprotinin blocks SARS-CoV-2 infection.  The authors claim that aprotinin inhibits the main SARS-CoV-2 cys-like protease, Mpro, an essential viral enzyme that processes the single virus polypeptide generated after virus cell entry.  These findings suggest a therapeutic use of aprotinin to reduce the viral load and the severity of COVID-19.  Nonetheless, although the aprotinin inhibition of CoV infection reported is of significance, I cannot consider docking a valid method to prove aprotinin binding to Mpro.  In addition, aprotinin is a serine protease inhibitor, whereas Mpro is a cysteine protease.  Therefore, the manuscript needs to incorporate biochemical or structural data that validate the inhibitor target, Mpro in particular. Therefore, I recommend a major revision of the manuscript based on the suggestions included below.

  1. It is necessary to define the aprotinin target. Aprotinin inhibition of Mpro must be studied with purified protein following described biochemical assays.  Alternatively, binding of aprotinin to the Mpro active site could be assessed by complex structure determination (X-ray or NMR).
  2. The introduction is very short and it does not present the topic of the manuscript to the reader. It should describe briefly the involvement of proteases in SARS-CoV-2 cell entry, the role of Mpro in CoV cell cycle, as well as previously described inhibitors of Mpro and entry proteases.
  3. As described in the Method section, the CPE was determined visually, which is not a quantitative way to assess infection. The authors should measure cell survival by OD after cell staining with crystal violet, and subsequently determine CPE.
  4. In Figure 3, are the period and conditions sufficient to mediate virus entry prior to addition of aprotinin? Temperature is not described. The SARS-CoV-2 entry pathway is more complex than a simple “adsorption”, requires time and it varies with cell type and temperature.  Authors should confirm that under the “adsorption” conditions the virus was able to penetrate into the cell before drug addition.
  5. Discussion, lane 281: Numerous SARS-CoV Mpro inhibitors have been described.

Author Response

Introductory note by the authors:

This is an unusual revision, since reviewer 2 asked us to perform additional time-of-addition experiments with a read-out after eight hours, a time point that roughly reflects one round of replication. Unexpectedly for us, these experiments revealed that entry inhibition makes a much more pronounced contribution to the anti-SARS-CoV-2 effects of aprotinin than we had originally anticipated. Hence, we had to revise our manuscript to an extent that makes it impossible to highlight the changes in a reasonable manner. However, we added a substantial amount of extra data and feel that this has substantially improved the manuscript. Please find the point-by-point responses to your requests below.

The manuscript shows that the serine protease inhibitor aprotinin blocks SARS-CoV-2 infection.  The authors claim that aprotinin inhibits the main SARS-CoV-2 cys-like protease, Mpro, an essential viral enzyme that processes the single virus polypeptide generated after virus cell entry.  These findings suggest a therapeutic use of aprotinin to reduce the viral load and the severity of COVID-19.  Nonetheless, although the aprotinin inhibition of CoV infection reported is of significance, I cannot consider docking a valid method to prove aprotinin binding to Mpro.  In addition, aprotinin is a serine protease inhibitor, whereas Mpro is a cysteine protease.  Therefore, the manuscript needs to incorporate biochemical or structural data that validate the inhibitor target, Mpro in particular. Therefore, I recommend a major revision of the manuscript based on the suggestions included below.

It is necessary to define the aprotinin target. Aprotinin inhibition of Mpro must be studied with purified protein following described biochemical assays.  Alternatively, binding of aprotinin to the Mpro active site could be assessed by complex structure determination (X-ray or NMR).

The introduction is very short and it does not present the topic of the manuscript to the reader. It should describe briefly the involvement of proteases in SARS-CoV-2 cell entry, the role of Mpro in CoV cell cycle, as well as previously described inhibitors of Mpro and entry proteases.

Authors response:

We agree that the docking results are too speculative and removed them. Moreover, our additional time-of addition experiments (Figure 3A) indicated that entry inhibition is a main anti-SARS-CoV-2 mechanism of action of aprotinin. Hence, other mechanisms are of limited relevance.

As described in the Method section, the CPE was determined visually, which is not a quantitative way to assess infection. The authors should measure cell survival by OD after cell staining with crystal violet, and subsequently determine CPE.

Authors response:

We know have quantified the antiviral activity by inhibition of CPE formation, immunostaining for the SARS-CoV-2 S protein, and inhibition of SARS-CoV-2-induced caspase activation. All three approaches resulted in similar results and are summarised in the novel Table 1 (line 202).

In Figure 3, are the period and conditions sufficient to mediate virus entry prior to addition of aprotinin? Temperature is not described. The SARS-CoV-2 entry pathway is more complex than a simple “adsorption”, requires time and it varies with cell type and temperature.  Authors should confirm that under the “adsorption” conditions the virus was able to penetrate into the cell before drug addition.

Authors response:

As mentioned above, we have performed additional time-of-addition experiments in an 8h format, which roughly reflects one round of replication. The results indicated that entry inhibition is much more important for the anti-SARS-CoV-2 activity of aprotinin than we initially had anticipated.

Discussion, lane 281: Numerous SARS-CoV Mpro inhibitors have been described.

Authors response:

As previously mentioned, we have removed all speculations about this.

Round 2

Reviewer 2 Report

Bojkova et al presented a revised manuscript showing that aprotonin can inhibit SARS-CoV-2 replication. As there have been other publications on this topic, this is not a novel study but confirms findings found by others. The paper has been slightly improved from its original form but still feels rushed and as if the authors have not spent sufficient time to describe their rationale, methods, and results.

Major concerns:

  1. The introduction is so short that it does not introduce the reader to the virus entry process, the viral S protein and the proteases required for virus entry. Additionally, there is no description of the protease inhibitors used in the study and what is currently known in the literature about protease inhibitors of SARS or other respiratory viruses. The intro is too short and needs to be expanded to include more key information required for the manuscript.
  2. A new caspase inhibitor assay was added to the manuscript; however, the rational, experimental set-up and description of the results was so short that it was hard to understand the reason for including this new experiment. The authors should expand the explanation of this experiment.
  3. What is table 1? There is no description or reference to it in the text.
  4. What does 5uM Aprotonin have a larger effect than 20uM in Figure 3? The data in this figure seem very inconsistent and it is hard to draw a conclusion about the effects of the Aprotonin on a one round of infection.
  5. The discussion should be expanded and discuss further studies that have been performed on SARS-CoV-2 in respect to protease inhibitors.

Minor comments:

1. Line 224 should read 1C.

2. Line 225 : Are the author sure of the values they have listed? While the values of 900 and 237 seem to be reasonable from the graph – the value of 54 seems to be wrong. The FFM6 decrease looks like the FFM1 which they claim has a value of 900. This should be checked.

Author Response

Major concerns:

Authors’ note: All changes in the manuscript are highlighted in yellow.

1) The introduction is so short that it does not introduce the reader to the virus entry process, the viral S protein and the proteases required for virus entry. Additionally, there is no description of the protease inhibitors used in the study and what is currently known in the literature about protease inhibitors of SARS or other respiratory viruses. The intro is too short and needs to be expanded to include more key information required for the manuscript.

Author response:

The Introduction was substantially amended to address this criticism. The additional paragraphs read now (lines 51-63):

“Cell entry of coronaviruses is mediated by interaction of the viral spike (S) protein with their host cell receptors, which differ between different coronaviruses [8]. For example, Middle East respiratory syndrome coronavirus (MERS-CoV) uses dipeptidyl peptidase 4 (DPP4) as cellular receptor [8]. Host cell entry of SARS-CoV-2 and of the closely related severe acute respiratory syndrome virus (SARS-CoV) is mediated by angiotensin-converting enzyme 2 (ACE2) [8-10]. S binding to ACE2 depends on S cleavage at three sites (S1, S2, and S2') by host cell proteases, typically by the transmembrane serine protease 2 (TMPRSS2), and can be inhibited by serine protease inhibitors [9,10]. Camostat was the first serine protease inhibitor that was shown to inhibit TMPRSS2 [9]. Subsequently, additional TMPRSS2 inhibitors including nafamostat and arbidol derivatives were demonstrated to interfere with SARS-CoV-2 internalization into host cells [11-13].

Aprotinin is a serine protease inhibitor, which has previously been shown to inhibit TMPRSS2 and suggested as treatment option for influenza viruses and coronaviruses [14,15]. Here, we investigated the effects of aprotinin against SARS-CoV-2.”

2) A new caspase inhibitor assay was added to the manuscript; however, the rational, experimental set-up and description of the results was so short that it was hard to understand the reason for including this new experiment. The authors should expand the explanation of this experiment.

Author response:

This additional assay format was introduced as an additional quantitative assay that determines the antiviral effects. Caspase 3/7 activation is in this case not an apoptosis assay but a quantitative read-out of the antiviral effects. We have explained this in more detail in the revised version (lines 214-221):

3.2 Quantification of the antiviral effects of aprotinin by measuring SARS-CoV-2-induced caspase 3/7 activation

Different viruses including SARS-CoV-2 have been shown to induce caspase 3 activation [29-32] and virus-induced caspase 3 activation has been used as read-out in assays that quantify the antiviral effects of drug candidates [31]. Hence, we used the Caspase-Glo® 3/7 Assay (Promega) as an additional quantitative method to determine the anti-SARS-CoV-2 activity of aprotinin. The results confirmed those obtained by CPE formation and S expression resulting in similar IC50 values (Figure 2, Table 1).”

3) What is table 1? There is no description or reference to it in the text.

Author response:

Table 1 presents the IC50 values of the anti-SARS-CoV-2 effects of aprotinin determined by three different approaches (CPE formation, S staining, caspase 3/7 activation) together to enable the direct comparison. The Table is referred to in line 187 and in line 221.

4) What does 5uM Aprotonin have a larger effect than 20uM in Figure 3? The data in this figure seem very inconsistent and it is hard to draw a conclusion about the effects of the Aprotonin on a one round of infection.

Author response:

Aprotinin does not exert a significant effect on SARS-CoV-2 replication in Figure 3A. The differences are within the noise. This is why there is no concentration-dependence. In contrast, remdesivir causes significant concentration-dependent effects. We have added a statistical analysis (one-way ANOVA and Dunnett’s multiple comparison test) to clarify this (see Figure 3A and Figure 3 legend, lines 239-240).

5) The discussion should be expanded and discuss further studies that have been performed on SARS-CoV-2 in respect to protease inhibitors.

Author response:

We have amended the Discussion as follows to address this comment (lines 304 to 307):

“Our findings are also in agreement with studies that reported other TMPRSS2 inhibitors to inhibit SARS-CoV-2 entry and replication [11-13]. In addition, furin has been shown to cleave and activate SARS-CoV-2 S and furin inhibitors have been demonstrated to exert anti-SARS-CoV-2 effects [38].”

Minor comments:

  1. Line 224 should read 1C.

Author response:

This was corrected (now line 188).

  1. Line 225 : Are the author sure of the values they have listed? While the values of 900 and 237 seem to be reasonable from the graph – the value of 54 seems to be wrong. The FFM6 decrease looks like the FFM1 which they claim has a value of 900. This should be checked.

Author response:

Many thanks. Yes, this was an error. The correct value is 584-fold. This was corrected (now line 189).

Reviewer 3 Report

The authors carried out a major revision of the manuscript, which changed substantially and it is now providing a distinct message. The current version is showing that aprotinin inhibits the SARS-CoV-2 cell entry, likely inhibiting the serine proteases that prime the S protein for virus-cell fusion. The authors used several methodologies to prove that aprotinin is blocking virus cell entry and infection. Quantitative assays are included, although the aprotinin's target is undefined. The introduction remains very short. As I stated in the previous revision, it should describe the proteases required for CoV cell entry and their roles in S protein processing.

Author Response

The authors carried out a major revision of the manuscript, which changed substantially and it is now providing a distinct message. The current version is showing that aprotinin inhibits the SARS-CoV-2 cell entry, likely inhibiting the serine proteases that prime the S protein for virus-cell fusion. The authors used several methodologies to prove that aprotinin is blocking virus cell entry and infection. Quantitative assays are included, although the aprotinin's target is undefined. The introduction remains very short. As I stated in the previous revision, it should describe the proteases required for CoV cell entry and their roles in S protein processing.

Author response:

The Introduction has been amended as follows to address this comment (lines 51-63, highlighted in yellow in the manuscript):

“Cell entry of coronaviruses is mediated by interaction of the viral spike (S) protein with their host cell receptors, which differ between different coronaviruses [8]. For example, Middle East respiratory syndrome coronavirus (MERS-CoV) uses dipeptidyl peptidase 4 (DPP4) as cellular receptor [8]. Host cell entry of SARS-CoV-2 and of the closely related severe acute respiratory syndrome virus (SARS-CoV) is mediated by angiotensin-converting enzyme 2 (ACE2) [8-10]. S binding to ACE2 depends on S cleavage at three sites (S1, S2, and S2') by host cell proteases, typically by the transmembrane serine protease 2 (TMPRSS2), and can be inhibited by serine protease inhibitors [9,10]. Camostat was the first serine protease inhibitor that was shown to inhibit TMPRSS2 [9]. Subsequently, additional TMPRSS2 inhibitors including nafamostat and arbidol derivatives were demonstrated to interfere with SARS-CoV-2 internalization into host cells [11-13].

Aprotinin is a serine protease inhibitor, which has previously been shown to inhibit TMPRSS2 and suggested as treatment option for influenza viruses and coronaviruses [14,15]. Here, we investigated the effects of aprotinin against SARS-CoV-2.”